# The relationship between maternal uterine artery doppler ultrasound during the second and third trimesters and infant cognitive development at one year of age: Findings from a South African prospective cohort study

Sydni A. Weissgold[1‡¤], Noha Ibrahim[2], Lucy Brink[3], Amy J. Elliott[4], Eva Loth[2], Vaheshta Sethna[1‡*], Hein Odendaal[3‡]

1 Social, Genetic & Developmental Psychiatry Centre, Institute of Psychiatry, Psychology & Neuroscience, Kings College London, United Kingdom, 2 Department of Forensic and Neurodevelopmental Sciences, Institute of Psychiatry, Psychology & Neuroscience, Kings College London, London, United Kingdom, 3 Department of Obstetrics and Gynaecology, Stellenbosch University, Cape Town, South Africa, 4 Avera Research Institute, Sioux Falls, South Dakota & Deptartment of Pediatrics, University of South Dakota School of Medicine, Sioux Falls, South Dakota, United States of America

‡ SAW and NI share joint first authorship. VS and HO share joint last authorship.
¤ Patrick Wild Centre, Division of Psychiatry, Centre for Clinical Brain Sciences, University of Edinburgh, Edinburgh, United Kingdom
* Vaheshta.sethna@kcl.ac.uk

## Abstract

### Introduction

Increased resistance to uterine artery blood flow is an index for poor pregnancy outcomes. This study aimed to examine the association between maternal uterine artery pulsatility index and cognitive development in infants aged one year, and whether placental dysfunction moderates this relationship.

### Methods

This was a prospective cohort study in an economically deprived community in South Africa. 1297 pregnant women with singleton gestations and their term infants were assessed. uterine artery pulsatility index, assessed by Doppler ultrasound in the second and third trimesters of pregnancy were examined. Placental dysfunction is indicated by maternal vascular malperfusion and accelerated villous maturation of the placenta. The primary outcome was infant cognitive development, assessed by the composite score of the Mullen Scales of Early Learning, at one year of age.

### Results

Higher uterine artery pulsatility index was associated with lower cognitive scores, when adjusting for alcohol consumption and antenatal depression in the second trimester ($\beta = -0.086$, $p = 0.007$), explaining 5% of the variance in the model. There was

**Data availability statement:** De-identified data from the Safe Passage Study is available through NICHD's Data and Specimen Hub (DASH): https://dash.nichd.nih.gov/. DASH is a centralized resource that allows researchers to access data from various studies via a controlled-access mechanism.

**Funding:** NIAAA and NIDCD (U01 AA016501) The funders were involved in the design of the original SPS but played no role in the design of this manuscript.

**Competing interests:** The authors declare that no competing interests exist.

**Abbreviations:** UtA-PI: Uterine Artery Pulsatility Index; MVM: Maternal Vascular Malperfusion; APM: Accelerated Placental Maturation; GA: Gestational Age; MSEL: Mullens Scales of Early Learning; EPDS: Edinburgh Postnatal Depression Scale.

no evidence of moderation by maternal vascular malperfusion or accelerated villous maturation of the placenta.

## Conclusions

Abnormal uterine artery pulsatility indices during the second trimester is associated with cognitive development in infants.

## Introduction

Complications during pregnancy have been shown to influence offspring neurodevelopment [1]. Increased resistance to uterine artery blood flow (i.e., elevated uterine artery pulsatility index (UtA-PI) as assessed by Doppler ultrasound), and placental dysfunction (i.e., maternal vascular malperfusion (MVM), a group of utero-placental lesions) are indicators of pregnancy complications – including stillbirth [2], fetal growth restriction [3], and low birthweight [4–8]. However, studies concerning longer-term neurodevelopment remain scarce.

To our knowledge, only one study has examined a link between UtA-PI (second trimester) and cognitive development in children aged two to three years [9], and demonstrated that resistance to blood flow in the uterine artery was not a risk factor for neurodevelopment. Probable biases in participant selection, classification of UtA-PI groups, loss to follow-up of participants with poor gestational outcomes, and the inclusion of both preterm and term deliveries in the high-resistance group, could have influenced results [9].

Similarly, evidence linking pathologic lesions associated with placental dysfunction and infant offspring neurodevelopment is sparse and focuses on high-risk pregnancies [10,11]. For example, MVM is predictive of poor cognitive development at two years of age, albeit in offspring born very preterm [10].

Accelerated villous maturation of the placenta (AMP), a feature of MVM, is linked to offspring outcomes. Furthermore, AMP is among the most common individual features of MVM which associates with increased UtA-PI [12]. Preterm placentas showing AMP are associated with improved birth outcomes [13], while pointing towards lower scores in cognitive development in older preterm and term infants aged 14 months [14].

Current evidence on UtA-PI and placental dysfunction is largely based on high-risk samples from middle-to-high-income settings. Since UtA-PI differs according to ethnicity [15,16] and given the wide range of ethnic samples examined, existing evidence should be interpreted with caution. Moreover, factors known to impact cognitive development such as maternal stress [17], lifestyle-related risk factors [18,19], and socio-demographic characteristics [20] require consideration in conjunction with UtA-PI and placental dysfunction. It is important to explore UtA-PI with offspring cognitive development, not only in the second [9], but also in the third trimester given the links between both periods of pregnancy with offspring outcomes [21,22]. Lastly, offspring neurodevelopment, in association with uterine artery blood flow and placental dysfunction, has not been assessed in the first year of life [9–11].

We therefore investigate the associations between maternal UtA-PI during the second and third trimesters of pregnancy and offspring cognition at one year of age, and whether this association was moderated by placental dysfunction. We hypothesise that increased UtA-PI will be associated with poorer offspring cognition at one year of age and that maternal malperfusion and accelerated villous maturation will moderate this relationship.

## Materials and methods

### Participants

A community-based cohort was recruited in Cape Town South Africa, as a subset of the Safe PASSage Study of the Prenatal Alcohol in Sudden Infant Death Syndrome and Stillbirth Network (N = 7060) [23]. The recruitment period for the Safe PASSage Study began on 6 August 2007 and was completed on 15 January 2015; the infants were followed up through August 2016. 1297 women were included in this 'embedded' study, with additional measures during pregnancy and outcome, for meeting the following inclusion criteria: 1) enrolled between gestational ages (GA of six weeks' zero days and 23 weeks' six days and assigned to the sub-study, 2) provided informed written consent in English or Afrikaans to participant in the Safe PASSage Study as well as for: infant follow-up evaluations up to one year after birth, collection of placental tissue, and use of tissues for future studies (participants were made aware that they would withdraw from the study at any time), 3) returned one year after birth for offspring cognitive assessments, 4) had a singleton pregnancy, 5) were 16 years or older, 6) completed at least one Doppler ultrasound assessment during pregnancy, and 7) enrolled in the Safe PASSage Study for the first time. Women were excluded if they met any of the following exclusion criteria: 1) were planning to have an abortion, 2) were planning to move from the catchment area before delivery, 3) received medical advice against study participation if, for example,a prospective participant required additional medical care, and 4) if they delivered preterm (<37 weeks' gestation) (Fig 1) [23,24].

### Measures

**Second and third trimester UtA-PI (exposure).** Doppler ultrasound scans were collected at two time points during pregnancy, at 20–24 weeks' and at 34–38 weeks' gestation, and analysed separately. UtA-PI data from 276 and 256 participants were missing from the first or second trimester, respectively. GA was estimated based on fetal biometry data

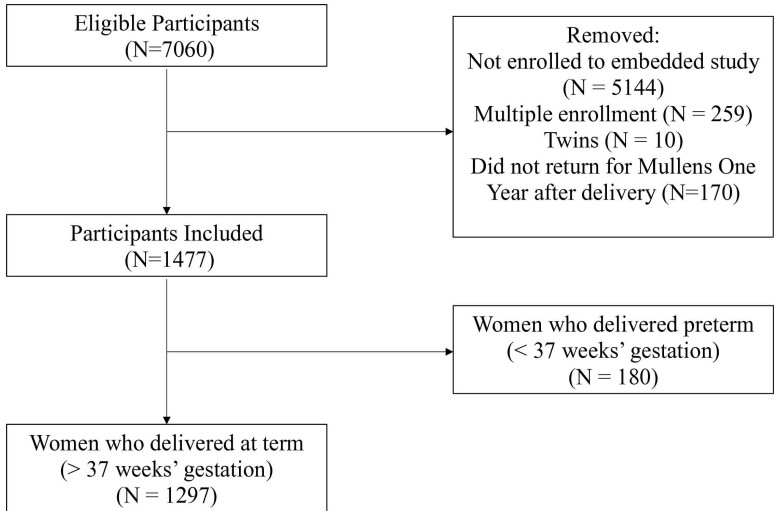

**Fig 1. Longitudinal flow of participants through the study.**

from the earliest ultrasound scans, performed by trained sonologists [23,25]. Careful transabdominal placement allows for ultrasound examination of uterine artery flow parameters [26]. The pulsatility index was calculated as the (maximum velocity – minimum velocity)/mean velocity [5]. Mean pulsatility index is used in all subsequent analyses. A higher pulsatility index indicates increased dysfunction.

**Placental dysfunction (potential moderators):** Placentas were collected at delivery, and subsequently fixed in 10% buffered formalin for transfer to perinatal pathologists [18]. Placental sections were taken, and slides stained with eosin and haematoxylin [18]. The pathologists involved in this process were blind to clinical observations. Inter-rater reliability between pathologists was held at or above 90% agreement on individual placental samples which were randomly selected by the monitoring site. Two placental histopathological diagnoses were included in this study, 1) MVM, and 2) AMP. Diagnoses aligned with the Amsterdam Placental Workshop Consensus Statement [27].

*MVM:* MVM are a collection of pathological insults seen in maternal decidual vessels, part of the uterine artery vasculature [3]. MVM lesions in this study include: small placenta for GA, macroscopic pathological infarction, decidual arteriopathy, accelerated maturation of the placenta, increased syncytial knots, distal villous hypoplasia, increased perivillous fibrin, increased extravillous trophoblasts, and microscopic pathologic infarction. A diagnosis of MVM comprised two or more types of MVM lesions present. Data from 318 placentas were missing.

*AMP:* AMP are hypermature, hypoplastic, and slender terminal villi resembling the histology of term villi with considerably increased intervillous space as compensation of MVM [28–30]. A diagnosis of AMP was made if placental samples showed higher number of mature villi than appropriate for GA. Data from 313 placentas were missing.

**One year infant cognition (outcome):** Infant cognition was measured by the Mullen Scales of Early Learning (MSEL) that assesses five cognitive abilities: gross and fine motor, visual reception, receptive language, and expressive language [31]. Sub-scales are given a T-Score with a mean of 50 and a standard deviation of 10 [31]. The Early Learning composite score, utilised as the primary measure of cognitive development in this study, refers to the sum of MSEL sub-scales (excluding gross motor) adjusted for age and early learning score. The Early Learning composite score is standardised (Mean = 100, Standard Deviation = 15) [31]. MSEL data from 62 offspring was not recorded.

**Maternal education (confounder):** Maternal education, quantified as number of years of formal education (primary education = years 0–7, high school = years 8–12, College/ University = Year 13 and above) completed, is assessed during the recruitment interview questionnaire. This is collected at time of study enrolment, prior to the first foetal visit; data is missing from two women.

**Lifestyle-related risk factor: alcohol consumption during pregnancy (confounder):** Using a modified Timeline Follow-Back assessment, women provided alcohol consumption history (including number and size of drinks per day, type of alcohol, and whether the drink included ice or was shared) for 30 days prior to each prenatal visit, two weeks' before and after conception, and 30 days after their last drinking day [23,25]. One standard drink (14 g of pure alcohol) was defined using the National Institute on Alcohol and Abuse and Alcoholism criteria [32]. Alcohol consumption data was collected from all women.

**Offspring variables: GA at delivery and birthweight Z-score (confounder):** Foetal biometry data was collected from Doppler ultrasounds. Gestational age was estimated based on earliest ultrasound biometry data [23]. Birthweight Z-Score is the international standard for comparing birthweight; gender, gestational age at birth and birthweight are adjusted for.

   **Statistical analysis.** Data was analysed using IBM SPSS Statistics Version 27.0 [33]. First, descriptive analyses (mean, standard deviation) were conducted to describe the exposure (maternal UtA-PI), outcome (offspring cognition), links between exposure and outcome, and the association between maternal UtA-PI at the second and third trimesters of pregnancy. Additionally, the relationship between potential covariates and study variables and UtA-PI during the second and third trimesters of pregnancy and MSEL composite scores were determined via Spearman's correlation. Potential covariates included maternal factors (alcohol consumption during pregnancy, education, depressive symptoms, hypertension and caesarean section) and offspring factors (GA at delivery and birthweight Z-score). Where a significant (i.e., $p < 0.05$) relationship was observed that variable was adjusted in subsequent multivariate.

Second, to analyse attrition, a one-way Analysis of Variance, or its non-parametric counterpart, Kruskal-Wallis, was conducted on sociodemographic data (age, education, income, upper arm circumference, employment, alcohol consumption during pregnancy) on those who completed both ultrasound scans (second and third trimester), and those who only completed one scan (second or third trimester) and were presumed to have dropped out or suffered fetal loss. Third, inferential analyses included two separate unadjusted simple linear regressions to examine whether maternal UtA-PI, during the second and third trimesters respectively, predicts one-year offspring MSEL performance (composite score and MSEL subscales). Next, using multiple linear regressions we tested whether maternal UtA-PI, at, at each time point during pregnancy, predicts each of the MSEL scores at one year of age after controlling for potential confounders.

Finally, we used the macro process tool to investigate whether the interaction term between maternal UtA-PI and offspring cognitive development was significantly moderated by placental dysfunction, namely MVM and AMP [34]. Offspring cognition was measured by the MSEL composite variable as a score of overall cognitive ability. PROCESS applies bootstrapping intervals to correct for bias; 95% bias-corrected bootstrapping levels were held at the default of 5000 [34]. PROCESS also provides the conditional effects of a dichotomous moderator (0 = no histopathological diagnosis given, 1 = diagnosis of placental histopathological given) if an interaction term is statistically significant.

**Ethics.** The original Safe Passage Study protocol and all subsequent modifications and addendums were approved by the Health Research Ethics Committee of Stellenbosch University (Reference N06/10/210). Participants were asked, as part of the original written informed consent, to approve the use of their data for future studies. Most participants approved, and the data of those who did not approve, are not used for any study.

## Results

Table 1 gives descriptive analyses of maternal sociodemographic characteristics and social stressors, and offspring outcomes. Participant's (N = 1297) age at enrolment ranged from 16 to 43 years (mean = 25 years, standard deviation = 5.80 years), and 48.0% of offspring were male. 99.8% of the population self-indicated mixed descent; the remaining 0.2% were black. Mean age of offspring at the one-year postnatal visit was 367.7 days (standard deviation = 12.3 days) (Table 1). Women who only completed the second trimester ultrasound, in comparison to those who returned for both scans, did not significantly differ in sociodemographic criteria. UtA-PI between the second and third trimester Doppler scans were strongly correlated (r = 0.71, p < 0.001).

Significant correlations of small effect sizes were observed. MSEL composite scores (outcome) were significantly correlated to maternal education (r = 0.103; p < 0.001), maternal alcohol consumption during pregnancy (r = −0.060, p = 0.036), GA at delivery (r = 0.129, p < 0.001) and birthweight Z-score (r = 0.080, p = 0.005). Hence, maternal education, maternal alcohol consumption during pregnancy, GA at delivery, and birthweight Z-score were used as confounders in subsequent multivariate analyses. In contrast, maternal variables, including hypertension (r = −0.028, p = 0.334), depressive symptoms (r = −0.019; p = 0.501) and caesarean section (r = 0.013, p = 0.684) were not associated with MSEL composite score at the set threshold.

A significant negative association of small effect size (d = −0.167, 95% CI [−0.26, −0.08]) was observed between second trimester UtA-PI and offspring MSEL composite score (β = −0.083, p = 0.010), which was largely unchanged when adjusting for covariates (β = −0.063, p = 0.051) (d = −0.126 (95% CI [−0.022, −0.04])) and explained 5% of the variance in the model, implying increased second trimester UtA-PI was associated with a lower early learning composite score (Fig 2). Individual MSEL subscale analyses are presented in Table 2. Furthermore, there was no evidence of moderation by either MVM or AMP on the association between second trimester UtA-PI and MSEL composite score, given the interaction terms (coef = 0.243, coef = 0.281, respectively) did not reach significance (p = 0.623, p = 0.597, respectively).

Similarly, significant negative association of small effect size (d = −0.148 (95% CI [−0.24, −0.06])) was observed between third trimester UtA-PI and MSEL composite score (β = −0.074, p = 0.020) (Fig 2). However, this association attenuated when adjustments were made for covariates ((β = −0.051, p = 0.108) (d = −0.102 (95% CI [−0.19, −0.01])).

**Table 1. Descriptive statistics of maternal characteristics, along with offspring characteristics in utero, at birth, and at one year of age.**

### Maternal Demographics

| | N | | Mean | | SD | | Minimum | | Maximum |
|---|---|---|---|---|---|---|---|---|---|
| Maternal Age (years) | 1297 | | 24.7 | | 5.8 | | 16.0 | | 43.0 |
| Maternal Mid-Upper Arm Circumference (mm)[a] | 1273 | | 278.0 | | 47.3 | | 190.0 | | 475.5 |
| Parity | 1296 | | 0.9 | | 1.1 | | 0.0 | | 6.0 |
| Maternal Alcoholic Drinks per Pregnancy[b] | 1297 | | 12.2 | | 28.0 | | 0.0 | | 389.5 |
| Income (ZAR)[c] | 945 | | 934.70 | | 611.04 | | 50.0 | | 6000.0 |
| | No | | Yes | | Total | | Missing | | – |
| | N | % | N | % | N | % | N | % | – |
| Edinburgh Referable Depression Score[e] | 632 | 48.7 | 665 | 51.3 | 1297 | 100 | 0 | 0 | – |
| Employment | 711 | 54.8 | 434 | 62.1 | 1145 | 88.3 | 152 | 13.2 | – |
| Ethnicity | | | | | | | | | |
| | N | | % | | Total | | Missing | | – |
| Cape Coloured or Mixed | 1295 | | 99.8 | | N | % | N | | – |
| Black or African-American | 2 | | 0.2 | | 1297 | 100 | – | % | – |

### Offspring Outcomes

| | N | | Mean | | SD | | Minimum | | Maximum |
|---|---|---|---|---|---|---|---|---|---|
| **Foetus In Utero (20–24 weeks')** | | | | | | | | | |
| Estimated Weight (g) | 849 | | 526.3 | | 86.7 | | 293.5 | | 885.6 |
| Foetal Head Circumference (mm) | 1011 | | 202.6 | | 12.2 | | 163.5 | | 237.8 |
| **Foetus In Utero (34–38 weeks')** | | | | | | | | | |
| Estimated Weight (g) | 683 | | 2409.5 | | 292.1 | | 524.7 | | 3963.6 |
| Foetal Head Circumference (mm) | 903 | | 311.3 | | 11.6 | | 210.9 | | 350.3 |
| **Child at Birth** | | | | | | | | | |
| Birthweight Z-Score[f] | 1290 | | −0.3 | | 1.0 | | −3.6 | | 2.8 |
| Gestational age at delivery (days) | 1297 | | 276.6 | | 8.5 | | 259.0 | | 301.0 |
| Sex | M | | Total | | Missing | | – | | – |
| | N | % | N | % | N | % | – | | – |
| | 623 | 48.0 | 1297 | 100.0 | | | | | |
| **Child at One Year** | | | | | | | | | |
| | N | | Mean | | SD | | Minimum | | Maximum |
| Infant Age at One Year (days) | 1288 | | 367.7 | | 12.3 | | 330.0 | | 400.0 |
| Ponderal Index[e] | 1273 | | 23.6 | | 2.2 | | 16.4 | | 34.2 |
| Head Circumference (mm) | 1291 | | 46.1 | | 1.4 | | 41.1 | | 54.7 |

[a]Upper Arm Circumference is used as an index of maternal weight.

[b]1 Drink = 14g of pure alcohol.

[c]1 USD = 18.76 ZAR.

[d]Edinburgh Postnatal Depression Scale is validated; referable index indicates if participant indicated risk of self-harm *or* scored >13.

[e]Birthweight Z-Score is the international standard for comparing birthweight; gender, gestational age at birth and birthweight are adjusted for.

[f]Ponderal Index calculated using formula $100 * weight (g)/ height^3 (cm)$.

N = Sample Size.

SD = Standard Deviation.

$p < 0.05$.

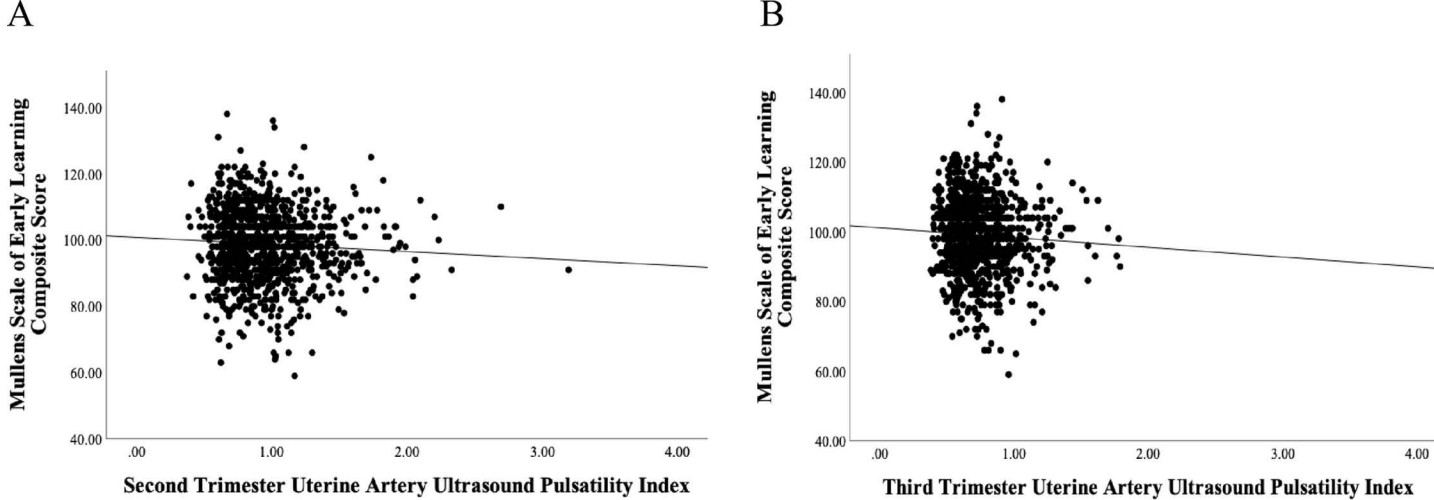

**Fig 2. Scatterplot of Uterine Artery Pulsatility Index, during the second.** (A) and third (B) trimesters, against offspring cognitive score at one year of age, measured by the Mullens Scale of Early Learning Composite score (corrected for early learning score and age).

**Table 2. Prediction of offspring cognitive development at 1 year by uterine artery pulsatility index in the second and third trimesters of pregnancy.**

|  |  |  |  |  |  | Unadjusted |  |  |  | Adjusted (Alcohol Consumption, Maternal Education, Gestational Age at Delivery, Birthweight Z-Score[*]) |  |  |  |
|---|---|---|---|---|---|---|---|---|---|---|---|---|---|
|  |  |  |  |  |  | Second Trimester |  | Third Trimester |  | Second Trimester |  | Third Trimester |  |
|  | N | Mean | SD | Minimum | Maximum | β | p | β | p | β | p | β | p |
| Gross Motor | 1290 | 51.21 | 13.43 | 20.0 | 80.0 | −0.05 | 0.147 | 0.01 | 0.754 | −0.02 | 0.474 | 0.02 | 0.564 |
| Fine Motor | 1277 | 49.20 | 9.08 | 22.0 | 77.0 | **−0.08** | **0.008** | **−0.09** | **0.003** | **−0.07** | **0.024** | **−0.77** | **0.015** |
| Visual Reception | 1271 | 48.87 | 7.42 | 20.0 | 80.0 | **−0.07** | **0.016** | **−0.08** | **0.017** | **−0.07** | **0.040** | **−0.06** | **0.044** |
| Receptive Language | 1288 | 49.73 | 5.99 | 22.0 | 74.0 | −0.03 | 0.310 | −0.00 | 0.914 | −0.01 | 0.808 | 0.02 | 0.602 |
| Expressive Language | 1284 | 50.01 | 5.97 | 20.0 | 71.0 | −0.03 | 0.414 | −0.02 | 0.532 | −0.01 | 0.681 | −0.01 | 0.881 |
| Composite[a] | 1235 | 99.12 | 10.28 | 59.0 | 138.0 | **−0.08** | **0.010** | **−0.07** | **0.020** | **−0.06** | **0.051** | −0.05 | 0.108 |

[a]Composite is a corrected sum of fine motor, visual reception, receptive language, expressive language (corrected for age and early learning score).

[*]Ponderal Index calculated using formula $100 * weight (g)/ height^3 (cm)$

N = Sample Size

SD = Standard Deviation

p < 0.05

See Table 2 for individual MSEL subscale analyses. Lastly, there was no evidence of moderation by MVM or AMP (coef = 0.224, p = 0.636, coef = 1.061, p = 0.303, respectively).

## Discussion

### Main findings

Our prospective longitudinal study of 1297 South African women is the first to demonstrate an association between increased resistance to uterine blood flow separately in the second trimester of pregnancy and infant cognition at one

year of age in a cohort of term births. Higher maternal UtA-PI, measured in the second and third trimesters in pregnancy, was associated with lower infant cognitive scores in unadjusted analyses. After adjusting for potential covariates, the association between uterine blood flow in the second trimester and offspring cognition at one year of age remained, however, the relationship between uterine artery blood flow in the third trimester and offspring cognition was not statistically significant. There was no evidence of moderation by placental dysfunction on the relationship between UtA-PI and offspring cognition.

### Strengths, limitations, and future directions

The large South African sample size provides a unique geocultural angle in our study. Our results, therefore, demonstrate robust findings which can direct future allocations of maternal health care resources for South African women in poor socioeconomic areas. In addition, the prospective design of our study allows for a deeper understanding of the temporal sequence of events during gestation that contribute to offspring cognition. Our study is unique in that we examine resistance to uterine blood flow and placental dysfunction in term, uncomplicated pregnancies; this is not routine practise. There was some selective attrition towards those of lower socioeconomic backgrounds, however, we do not expect this to skew our results.

Recruitment bias is possible as a limited number of women were selected on recruitment days from those waiting for their first antenatal visits. Additionally, maternal depression and alcohol consumption data was collected via self-report surveys; this method is subject to biases such as recall and fear of stigma [23]. However, it is our experience, and also that of health care workers doing similar research in other local communities, that the women report honestly and accurately on their drinking habits [16]. Furthermore, our effect sizes were small, demonstrating that further research is needed to confirm our findings. Hence the current findings should be interpreted with caution. It is pertinent that future studies examine additional factors, such as environmental and biophysical factors, which may influence cognitive development. Furthermore, the intricacy of the relations between accelerated maturation and preterm births limits the interpretation of our results. Our removal of women who delivered preterm was selected to minimise bias, however, much of the effect could have been lost due to this removal. It would be interesting to conduct a study of similar design which stratifies participants by term delivery to examine how much of the effect observed is due to preterm birth. It is possible that placental dysfunction plays a role in offspring cognitive development; however, our sample size may be too small to demonstrate such. Future research with larger sample sizes is required to elucidate the associations between placental function and offspring cognitive development. Larger sample sizes will afford the findings with increased statistical power and provide the opportunity to include a truly generalisable sample. Lastly, it is suggested that future studies stratify the sample by gender.

### Interpretation

Our study demonstrates a small role for the maternal uterine vascular system in offspring cognitive development. However, the observed association between resistance to uterine blood flow and offspring cognitive development contrasts with existing evidence [9]. Okido et al. (2020) examined women with UtA-PI above and below the 90th centile in the second trimester with offspring cognition at two to three years of age using the Bayley's Scale of Infant Development-III. The sample in this study included both full and pre-term offspring, and maternal age, weight, racial origin and gestational age at birth were adjusted within analyses. It is thus likely that methodological differences influence findings. These include 1) the different cognitive measure used (i.e., MSEL versus Bayley's Scale of Infant Development-III), 2) the use of a continuous measurement scale of UtA-PI in our study, compared with a dichotomous scale in Okido's et al. (2020) study, 3) the difference in offspring age at assessment (assessed at one year in this study, compared with assessment at ages two to three in Okido et al. (2020), 4) the differences in sample sizes, and 5) the inclusion of preterm and full-term offspring in Okido's et al (2020) sample.

We demonstrate that resistance to uterine blood flow, in both trimesters, is marginally associated with MSEL sub-scales fine motor and visual reception (Table 2). Given that the composite MSEL score is an adjusted sum of the MSEL sub-scales, this result suggests that fine motor and visual reception scores could be a driving factor in the observed association. This may provide further information regarding underlying fetal programming processes [4]. Future research may investigate why only a subset of MSEL scales associate with UtA-PI. There was a difference in the association between maternal UtA-PI and offspring cognition based on time of ultrasound assessment. Higher maternal UtA-PI in the 2nd trimester was significantly associated with offspring cognition at one year; the same relationship was observed when examining maternal UtA-PI in the 3rd trimester, however, this association did not reach statistical significance, despite the stability of UtA-PI between the two time points [12]. This may be explained, in part, by a population study of 32,286 singleton births which demonstrated that maternal stress during weeks' 19–24 of gestation were most strongly associated with birth outcomes, such as low birth weight and small for gestational age [35]. Our results extend such findings via suggesting that maternal stress during the 2nd trimester may be temporally important for subsequent cognitive development in infancy. Additional investigations from the Safe PASSage Study demonstrate that 2nd trimester maternal alcohol consumption is associated with foetal growth [36]. This provides a link between confounding variables accounted for in this study, i.e., alcohol consumption, with the timing of exposure, i.e., the 2nd trimester, suggesting the importance of both maternal stressors and timing of foetal exposure in offspring development. While the mechanistic links between maternal stress, uterine blood flow and offspring cognitive development have yet to be determined, it is possible that the relationship between maternal uterine artery blood flow in the 2nd trimester and offspring cognitive function at one year of age may be related to hypoxic events, which are common byproducts of disrupted circulation [37]. Nevertheless, our study demonstrates the importance of resistance to uterine blood flow assessments during pregnancy for examinations of infant neurodevelopment.

MVM did not moderate the association between maternal UtA-PI and offspring cognition at one year of age. This contradicts findings by van Vliet et al. (2021), demonstrating that MVM was more strongly associated with offspring cognitive development at two years of age than chorioamnionitis, a placental lesion relating to infection [10]. Our study, in contrast, provides an independent examination of MVM's influence on offspring cognition. However, our study confirms findings from Gardella et al. (2021) where no link was found between maternal malperfusion and cognitive development in growth-restricted preterm offspring aged two years [11]. It is possible that the observed variation in findings between van Vliet et al (2021), Gardella et al. (2021), and our study, result from differential inclusion of placental lesions in the definition of MVM between studies, as well as samples consisting of high-risk births.

Similarly, to MVM, AMP did not moderate the relationship between maternal UtA-PI and offspring cognition. Therefore, our study does not support previous findings that AMP is associated with poorer cognitive outcomes in the first 14 months of life [14], nor does it confirm Christian & Grynspan's (2019) hypothesis that accelerated villous maturation is associated with improved birth outcomes [13]. Our moderation results may not have reached significance because preterm cases were excluded; this is explained by Brink's et al. (2022) demonstration of increased incidence of AMP in preterm births [38]. It is possible that AMP processes are solely instigated in response to triggers related to prematurity [38].

In clinical practise, obstetricians and maternal-foetal specialists should be informed of the possibility of developmental delay in children when abnormal uterine Doppler assessments had been recorded in the mother during the pregnancy, to ensure that these children are carefully assessed at appropriate times and remedial treatment instituted when necessary.

## Conclusion

Overall, our study demonstrated that in a South African cohort, abnormally high resistance to uterine blood flow during the second trimester of pregnancy is associated with poorer offspring cognitive performance at one year of age. This study failed to demonstrate that placental dysfunction moderates the relationship between UtA-PI and offspring cognitive development at one year of age. The results from our study contribute to our understanding of maternal gestational health and

how this may influence fetal neurodevelopment, as well as aid in the development of biomarker for early infant cognitive risk.

## Supporting information

**S1 File. PLOS questionnaire on inclusivity in global research.**
(DOCX)

**S2 File. Dataset.**
(XLSX)

## Acknowledgments

We would like to thank the mothers and children who participated in the study. We would also like to acknowledge the great support of the management and administrative personnel of the original Safe Passage Study and the pathologists and ultrasonologists of Tygerberg Hospital who performed the relevant examinations.

## Author contributions

**Conceptualization:** Sydni Weissgold, Noha Ibrahim, Lucy Brink, Amy J. Elliott, Vaheshta Sethna, Hein Odendaal.

**Data curation:** Lucy Brink, Amy J. Elliott, Hein Odendaal.

**Formal analysis:** Sydni Weissgold, Noha Ibrahim, Lucy Brink, Vaheshta Sethna, Hein Odendaal.

**Funding acquisition:** Hein Odendaal.

**Investigation:** Lucy Brink, Hein Odendaal.

**Methodology:** Lucy Brink, Hein Odendaal.

**Project administration:** Lucy Brink.

**Supervision:** Noha Ibrahim, Vaheshta Sethna, Hein Odendaal.

**Visualization:** Sydni Weissgold, Vaheshta Sethna.

**Writing – original draft:** Sydni Weissgold, Noha Ibrahim, Vaheshta Sethna.

**Writing – review & editing:** Sydni Weissgold, Noha Ibrahim, Lucy Brink, Amy J. Elliott, Eva Loth, Vaheshta Sethna, Hein Odendaal.

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
