## [Decision Letter · Decision Letter 0]

18 Feb 2025

PONE-D-24-55765The Relationship Between Maternal Uterine Artery Doppler Ultrasound during the Second and Third Trimesters and Infant Cognitive Development at One Year of Age: Findings from a South African Prospective Cohort StudyPLOS ONE

Dear Dr. Sethna,

Thank you for submitting your manuscript to PLOS ONE. After careful consideration, we feel that it has merit but does not fully meet PLOS ONE’s publication criteria as it currently stands. Therefore, we invite you to submit a revised version of the manuscript that addresses the points raised during the review process.

We look forward to receiving your revised manuscript.

Kind regards,

Mohammad Haddadi

Academic Editor

PLOS ONE

Journal Requirements:

3. Please include your tables as part of your main manuscript and remove the individual files. Please note that supplementary tables (should remain/ be uploaded) as separate "supporting information" files.

4. In the online submission form, you indicated that [Data can be made available through request to the corresponding author].

5. Thank you for stating the following financial disclosure: [NIAAA and NIDCD (U01 AA016501)].

6. Thank you for stating the following in your Competing Interests section: [No].

Additional Editor Comments:

The manuscript titled "The Relationship Between Maternal Uterine Artery Doppler Ultrasound During the Second and Third Trimesters and Infant Cognitive Development at One Year of Age: Findings from a South African Prospective Cohort Study" is well-written and clear. The study’s aim is compelling, and the methodology is appropriate. I enjoyed reading it.

To enhance the manuscript, I suggest the following revisions:

Please avoid using abbreviations in the abstract.

In the Ethics section, specify whether consent for participation and publication was obtained, given the prospective nature of the study.

Expanding on the clinical and research implications in the Discussion section would be beneficial.

Reviewers' comments:

Reviewer's Responses to Questions

**Comments to the Author**

1. Is the manuscript technically sound, and do the data support the conclusions?

Reviewer #1: Yes

Reviewer #2: Yes

2. Has the statistical analysis been performed appropriately and rigorously? 

Reviewer #1: Yes

Reviewer #2: Yes

3. Have the authors made all data underlying the findings in their manuscript fully available?

Reviewer #1: Yes

Reviewer #2: Yes

4. Is the manuscript presented in an intelligible fashion and written in standard English?

Reviewer #1: Yes

Reviewer #2: Yes

5. Review Comments to the Author

Reviewer #1: the manuscript is interesting despite not only UtA are correlated with neurodevelopmentall delay but also birth whicht, inasnuch the null hypothesis of the study has to be changed and highlight the outcomes better for reders

Reviewer #2: The authors have done an excellent job in writing this article. It is well-written, and the subject matter is highly relevant. However, two revisions are recommended. First, a paragraph discussing the clinical implications of the findings should be included. Second, further elaboration on suggestions for future research would strengthen the manuscript, particularly expanding on the potential for studies with larger sample sizes, which is briefly mentioned.

6. PLOS authors have the option to publish the peer review history of their article (what does this mean? ). If published, this will include your full peer review and any attached files.

**Do you want your identity to be public for this peer review?** For information about this choice, including consent withdrawal, please see our Privacy Policy .

Reviewer #1: **Yes: ** Erich Cosmi

Reviewer #2: No

---

## [Author Response · Author response to Decision Letter 1]

25 Mar 2025

Reviewers’ comments

We would like to thank the editors and reviewers for their helpful comments. Each point raised have been addressed below and as track changes on the manuscript as well.

Journal Requirements:

We have carefully reformatted the manuscript to align with PLOS ONE’s style requirements, ensuring compliance throughout. To the best of our knowledge, the only remaining formatting consideration is the reference style. We have formatted the references using PLOS ONE’s citation style in EndNote, which displays in-text reference numbers as superscripts. We hope this is acceptable, but we are happy to make any further adjustments as needed.

Thank you for your guidance.

We have completed the PLOS ONE questionnaire on inclusivity in global research to the best of our ability and where relevant. The completed form has been attached as supporting information in this revised submission.

3. Please include your tables as part of your main manuscript and remove the individual files. Please note that supplementary tables (should remain/ be uploaded) as separate "supporting information" files.

We have incorporated the two tables into the main manuscript as requested. Table 1 is located at line 140 and Table 2 is at line 269.

4. In the online submission form, you indicated that [Data can be made available through request to the corresponding author].

The complete data set of the full study will be available on the National Institute of Child Health and Human Development Data and Specimen Hub Data (NICHD DASH) with free access to the public.

The data is not available in the manuscript or as supplementary material but could be requested from the authors.

5. Thank you for stating the following financial disclosure: [NIAAA and NIDCD (U01 AA016501)].

We have added the following statement to the funding section on the title page:

“The funders were involved in the design of the original SPS. The funders had no role in the study design, data collection and analysis, decision to publish or preparation of the manuscript.”

Thank you for your guidance. Please let us know if any further revisions are required.

6. Thank you for stating the following in your Competing Interests section: [No].

We have included a statement of no competing interests in the cover letter attached to this revised submission. Please let us know if further revisions are needed.

We have included captions for Supporting Information files at the end of the manuscript (line 421).

The reference list has been reviewed and to the best of our knowledge is complete and correct. If there are any references that the editor believes should be removed, please do let us know.

Additional Editor Comments:

The manuscript titled "The Relationship Between Maternal Uterine Artery Doppler Ultrasound During the Second and Third Trimesters and Infant Cognitive Development at One Year of Age: Findings from a South African Prospective Cohort Study" is well-written and clear. The study’s aim is compelling, and the methodology is appropriate. I enjoyed reading it.

To enhance the manuscript, I suggest the following revisions:

Please avoid using abbreviations in the abstract.

We have revised the abstract to remove all abbreviations, ensuring clarity and alignment with journal guidelines.

In the Ethics section, specify whether consent for participation and publication was obtained, given the prospective nature of the study.

We have added the following statement to the Ethics section (lines 220-222):

“Participants provided written informed consent, including approval for the use of their data in future studies. Data from those who did not provide consent were not used”.

Expanding on the clinical and research implications in the Discussion section would be beneficial.

We appreciate this valuable suggestion and have expanded on the clinical implications in the Discussion section. Specifically, we have emphasized the importance of obstetricians and maternal-fetal specialists being aware of the potential for developmental delay in children with abnormal uterine Doppler findings during pregnancy. This awareness can help ensure timely assessments and early intervention where needed (Discussion, lines 390-93).

We also sincerely thank the editor for their time and consideration in reviewing our manuscript.

Reviewers' comments:

Reviewer's Responses to Questions

Comments to the Author

1. Is the manuscript technically sound, and do the data support the conclusions?

Reviewer #1: Yes

Reviewer #2: Yes

2. Has the statistical analysis been performed appropriately and rigorously?

Reviewer #1: Yes

Reviewer #2: Yes

3. Have the authors made all data underlying the findings in their manuscript fully available?

Reviewer #1: Yes

Reviewer #2: Yes

4. Is the manuscript presented in an intelligible fashion and written in standard English?

Reviewer #1: Yes

Reviewer #2: Yes

5. Review Comments to the Author

Reviewer #1: the manuscript is interesting despite not only UtA are correlated with neurodevelopmentall delay but also birth whicht, inasnuch the null hypothesis of the study has to be changed and highlight the outcomes better for reders

We sincerely appreciate your thoughtful feedback and for recognizing the value of our manuscript. Regarding your comment on the null hypothesis, we would like to clarify that our study hypothesis is explicitly stated in the final paragraph of the introduction.

Reviewer #2: The authors have done an excellent job in writing this article. It is well-written, and the subject matter is highly relevant. However, two revisions are recommended. First, a paragraph discussing the clinical implications of the findings should be included. Second, further elaboration on suggestions for future research would strengthen the manuscript, particularly expanding on the potential for studies with larger sample sizes, which is briefly mentioned.

We thank Reviewer #2 for their helpful revisions, which we agree will improve the quality of our manuscript. We have added a comment on the clinical implications of the findings: “Obstetricians and Maternal-Fetal specialists should be informed of the possibility of developmental delay in children when abnormal uterine Doppler assessments had been recorded in the mother during the pregnancy, to ensure that these children are carefully assessed at appropriate times and remedial treatment instituted when necessary.” (line 378-81)

We have further amended the title of the ‘Strengths and Limitations’ section within the Discussion to be ‘Strengths, Limitations and Future Directions’ to accurately reflect the content of this sub-section. The following points have been added to this section to reflect on future directions for research, with a focus on sample size of future studies:

- “It would be interesting to conduct a study of similar design which stratifies participants by term delivery to examine how much of the effect observed is due to preterm birth.” (line 305-6)

- “Future research with larger sample sizes is required to elucidate the associations between placental function and offspring cognitive development. Larger sample sizes will afford the findings with increased statistical power and provide the opportunity to include a truly generalisable sample.” (line 308-11)

6. PLOS authors have the option to publish the peer review history of their article (what does this mean?). If published, this will include your full peer review and any attached files.

Do you want your identity to be public for this peer review? For information about this choice, including consent withdrawal, please see our Privacy Policy.

Reviewer #1: Yes: Erich Cosmi

Reviewer #2: No

Thank you again for reviewing our manuscript and for the points raised.

---

## [Decision Letter · Decision Letter 1]

28 Mar 2025

The Relationship Between Maternal Uterine Artery Doppler Ultrasound during the Second and Third Trimesters and Infant Cognitive Development at One Year of Age: Findings from a South African Prospective Cohort Study

PONE-D-24-55765R1

Dear Dr. Sethna, 

We’re pleased to inform you that your manuscript has been judged scientifically suitable for publication and will be formally accepted for publication once it meets all outstanding technical requirements.

Kind regards,

Mohammad Haddadi

Academic Editor

PLOS ONE

Additional Editor Comments (optional):

Reviewers' comments:

Reviewer's Responses to Questions

**Comments to the Author**

1. If the authors have adequately addressed your comments raised in a previous round of review and you feel that this manuscript is now acceptable for publication, you may indicate that here to bypass the “Comments to the Author” section, enter your conflict of interest statement in the “Confidential to Editor” section, and submit your "Accept" recommendation.

Reviewer #2: All comments have been addressed

Reviewer #3: All comments have been addressed

2. Is the manuscript technically sound, and do the data support the conclusions?

Reviewer #2: Yes

Reviewer #3: Yes

3. Has the statistical analysis been performed appropriately and rigorously? 

Reviewer #2: Yes

Reviewer #3: Yes

4. Have the authors made all data underlying the findings in their manuscript fully available?

Reviewer #2: Yes

Reviewer #3: Yes

5. Is the manuscript presented in an intelligible fashion and written in standard English?

Reviewer #2: Yes

Reviewer #3: Yes

6. Review Comments to the Author

Reviewer #2: Thank you for your thoughtful revisions and detailed responses to the reviewer comments. After reviewing the updated manuscript, I find that all concerns have been adequately addressed, and the manuscript is now suitable for publication.

Reviewer #3: This paper presents a novel investigation into the association between umbilical artery Doppler findings and infant cognitive performance. The study employs a robust methodology, rigorous statistical analysis, and a clear reporting style. Given its strong scientific merit, I recommend acceptance for publication.

7. PLOS authors have the option to publish the peer review history of their article (what does this mean? ). If published, this will include your full peer review and any attached files.

**Do you want your identity to be public for this peer review?** For information about this choice, including consent withdrawal, please see our Privacy Policy .

Reviewer #2: No

Reviewer #3: No

---

## [Editor Report · Acceptance letter]

PONE-D-24-55765R1

PLOS ONE

Dear Dr. Sethna,

I'm pleased to inform you that your manuscript has been deemed suitable for publication in PLOS ONE. Congratulations! Your manuscript is now being handed over to our production team.

Kind regards,

on behalf of

Dr. Mohammad Haddadi

Academic Editor

PLOS ONE